# Airborne eDNA captures three decades of ecosystem biodiversity

Alexis R. Sullivan[1,2,8], Edvin Karlsson[1,3,8], Daniel Svensson[1], Björn Brindefalk[3,4], Jose Antonio Villegas[1], Amanda Mikko[5], Daniel Bellieny[1], Abu Bakar Siddique[1,6], Anna-Mia Johansson[7], Håkan Grahn[3], David Sundell[3], Anita Norman[2], Per-Anders Esseen[1], Andreas Sjödin[3], Navinder J. Singh[2], Tomas Brodin[2], Mats Forsman[3] & Per Stenberg[1,3]✉

Biodiversity loss threatens ecosystems and human well-being, making accurate, large-scale monitoring crucial. Environmental DNA (eDNA) has enabled species detection from substrates such as water, without the need for direct observation. Lately, airborne eDNA has been showing promise for tracking organisms from insects to mammals in terrestrial ecosystems. Conventional biodiversity assessments are often labor-intensive and limited in scope, leaving gaps in our understanding of ecosystem response to environmental change. Here, we demonstrate that airborne eDNA can detect organisms across the tree of life, quantify changes in abundance congruent with traditional monitoring, and reveal land-use induced regional decline of diversity in a northern boreal ecosystem over more than three decades. By analyzing 34 years of archived aerosol filters, we reconstruct weekly temporal relative abundance data for more than 2700 genera using non-targeted methods. This study provides unified, ecosystem-scale biodiversity surveillance spanning multiple decades, with data collected at weekly intervals on both the individual species and community level. Previously, large scale analyses of ecosystem changes, targeting all types of organisms, has been prohibitively expensive and difficult to attempt. Here, we present a way of holistically doing this type of analysis in a single framework.

Quantifying biodiversity is critical in the face of rapid environmental change[1,2], but traditional methods like visual surveys and trapping are often difficult, expensive, and slow. Environmental DNA (eDNA) offers a promising solution by enabling species detections from substrates like soil and water without direct observation[3]. In addition to their logistical benefits, eDNA methods can simultaneously be used to survey all domains of life, without inherent bias towards conspicuous species. As habitat loss accelerates, eDNA from archival substrates,

such as sediments, may prove valuable as records of irrevocably altered ecosystems[4].

eDNA-based monitoring methods have proved particularly successful in aquatic ecosystems[3]. Water disperses and homogenizes DNA from microbes to mammals and is relatively easy to sample with standard methods[3]. In contrast, sources of terrestrial eDNA, although valuable for targeted monitoring, are generally too rare, ephemeral, or heterogeneous to provide comprehensive biodiversity data[5]. While a

[1]Department of Ecology and Environmental Sciences, Umeå University, Umeå, Sweden. [2]Department of Wildlife, Fish and Environmental Studies, Swedish University of Agricultural Sciences, Umeå, Sweden. [3]CBRN Defence and Security, Swedish Defence Research Agency (FOI), Umeå, Sweden. [4]Department of Molecular Biosciences, The Wenner-Gren Institute, Stockholm University, Stockholm, Sweden. [5]Umeå Plant Science Centre, Department of Plant Physiology, Umeå University, Umeå, Sweden. [6]Department of Plant Biology, Swedish University of Agricultural Sciences, Uppsala, Sweden. [7]Department of Molecular Biology, Umeå University, Umeå, Sweden. [8]These authors contributed equally: Alexis R. Sullivan, Edvin Karlsson. ✉e-mail: Per.Stenberg@umu.se

single water sample can contain eDNA from an entire trophic network[6], capturing this breadth of diversity on land can entail collecting samples of soil, scat, vegetation, and bulk invertebrates[7].

Given these challenges, air has emerged as a terrestrial analog to water for biodiversity monitoring[8–10]. Air, like water, is an effective dispersal medium, and the near-surface atmosphere is full of particles from microbes, plants, fungi, and animals[11–14]. Airborne microbial communities are connected to their terrestrial sources[12,13] and vary in response to season[15,16] and landcover[13,15,17]. In the last five years, studies have also detected airborne eDNA from plants outside their flowering season[18] and from animals ranging from the volant to entirely earthbound[8–10,14,19–23].

Here, we apply shotgun sequencing to archived air filters and reconstruct microbe-to-mammal diversity over three decades. We develop an integrated framework to robustly assign sequences to organisms, delineate potential spatial sources of airborne eDNA, and reconstruct seasonal and long-term trends. This allows us to survey more than 2700 genera, recover vertebrate abundance indices congruent with traditional monitoring, and detect a decline in biodiversity consistent with contemporaneous forest management. Our results show airborne eDNA can monitor biodiversity and underscore the immense latent potential in the thousands of aerosol monitoring stations deployed worldwide[10].

## Results and discussion

We sequenced airborne eDNA adhered to archived air filters from a radionuclide monitoring station in the boreal forest of northern Sweden (67.84°N, 20.42°E, see Supplementary Methods). As part of the station's routine activities, surface-level air is continuously pumped ($1000\ m^3\ hr^{-1}$; $>100{,}000\ m^3\ week^{-1}$) through $0.2\ \mu m$ glass fiber filters, which are changed weekly and stored long-term in airtight containers. Airborne eDNA is extremely dilute ($0.44\text{-}8.60\ ng\ m^{-3\ 24}$) outside of seasonal pulses of spores and pollen, but the large volume sampled by the radionuclide station enabled us to recover enough eDNA for shotgun sequencing with minimal library amplification, which helps to mitigate biases introduced by PCR[25]. In total, we generated *ca.* 30 terabases from 380 weekly filters from non-winter weeks in even-numbered years from 1974 to 2008 (Supplementary Fig. 12).

### Organisms from all domains of life are accurately identified in airborne eDNA

Linking eDNA sequences to their source organisms is challenging, with errors potentially introduced at every stage from field collection to bioinformatic analysis. This can lead to improbable, if not impossible, species detections, like the duck-billed platypus (*Ornithorhynchus anatinus*) appearing in eDNA substrates worldwide[26]. While using laboratory methods robust to contamination (see Supplementary Methods and analysis of lab and field controls in Supplementary Fig. 20), we also detected the platypus and other unlikely taxa after optimizing a lowest common ancestor read classifier[27] (Supplementary Methods).

Rather than simply excluding unambiguous false positives, we used them to train a gradient boosting machine[28] (GBM) to more reliably distinguish spurious from valid detections. We defined 31 statistics summarizing read-level classifications and examined over 150,000 global species occurrences[29] to curate a negative ($n = 326$) and positive ($n = 279$) training dataset. Once trained, the GBM assigned probability scores for unknown genera, which were converted to binary classifications based on error rates for out-of-sample test data ($n = 91$; Supplementary Methods).

We identified 2739 high-confidence genera with an estimated 93% precision and 71% recall based on out-of-sample test data. We validated a subset of genera using alignment-based methods, including taxa with low detection probabilities from airborne eDNA in previous studies, such as fish, ungulates, amphibians, and birds[20,21]. Validation results

indicated 90% precision of the GBM and a low incidence (5.4%) of label noise in the training dataset (Supplementary Data 7). Genus-level classifications were most reliable for mammals, birds, and fish, followed by well-sequenced plant and fungal lineages (Supplementary Data 7).

Plants with wind-dispersed pollen or spores, flying insects, microbes, and macrofungi were abundant in our data, all of which are well-represented in the reference database and the local landscape. Biomass[30,31], population size[30,31], habitat[32,33], dispersal mechanisms[34], and tissue source[33,35] influence eDNA abundances in the environment, and detection probabilities are further modulated by study design, including sampling volume, sequencing effort, and reference database composition. While airborne eDNA concentrations can be very low[24], particularly for non-volant animals, combining deep sequencing with high-volume samples enabled the detection of organisms spanning 69 phyla, 173 classes and three domains, from both aquatic and terrestrial habitats (Fig. 1 and Supplementary Data 6).

### Bioaerosol catchments are quantifiable and stable

Atmospheric dispersion models estimate the geographic extent of airborne eDNA sources – or catchment areas – by simulating the transport, mixing, and deposition of particles under changing meteorological conditions (e.g., wind fields, precipitation, temperature; see Supplementary Methods). Airborne eDNA emitted within an estimated catchment area have the potential to be transported to the monitoring station, given historic weather data. However, detecting any specific organism depends on additional factors, including particle quantity and emission rates, aerosolization processes, and the residence time of retrievable DNA. Instead, catchment areas provide a physically realistic geographic context for interpreting changes in airborne eDNA over time.

Catchment areas are shaped by the aerodynamic properties of airborne particles, with diameter a key determinant of residence time, transport distance, and deposition rates. We estimated weekly catchment areas for 60, 22, and 5 $\mu m$ particles, representing a range of forest bioaerosols, including pollen[36,37], fungal spores[38,39], and bacterial aggregates[24,40,41]. On an annual basis, more than 50% of 60, 22, and 5 $\mu m$ particles originated within 20 km ($\pm 5.1$), 50 km ($\pm 17.7$), and 310 km ($\pm 38.4$), respectively (Supplementary Fig. 4). Catchment areas were elliptical (Fig. 2A) and similar in shape across particle sizes (Supplementary Table 1 and Supplementary Data 1). Interannual variation was minimal (Supplementary Table 2), but catchment areas for 22 and 5 $\mu m$ particles varied significantly between weeks (Supplementary Table 3). While more specific models require further research, the stability of the catchment areas suggests that long-term trends in airborne eDNA are likely to reflect changes within the same geographic region, rather than changes of the sampling area.

Dispersion models are powerful tools for simulating the transport and deposition of particles with specific aerodynamic properties. However, these properties are unknown for most bioaerosols, and the spatial scale of airborne eDNA is poorly constrained. In contrast, trajectory ensemble receptor models (TERMs) provide qualitative estimates of source regions using only temporal variation in eDNA abundances and a reconstruction of air mass history[42–45]. We applied TERMs to three taxa with unambiguous landscape-level distributions: moose (*Alces*), a forest-dwelling browser; cod (*Gadus*), a demersal marine fish; and reindeer (*Rangifer*), a seasonal migrant between forests and alpine pastures.

Cod eDNA relative abundances were highest when air masses arrived from the north, after passing over marine environments at least 160 km away from the monitoring station (Fig. 2B). In contrast, moose relative abundances increased when air masses arrived from the south and southeast, aligning with known population densities in the boreal forest[46,47] (Fig. 2C). Geographic source estimates for reindeer eDNA varied seasonally, consistent with their annual migration between

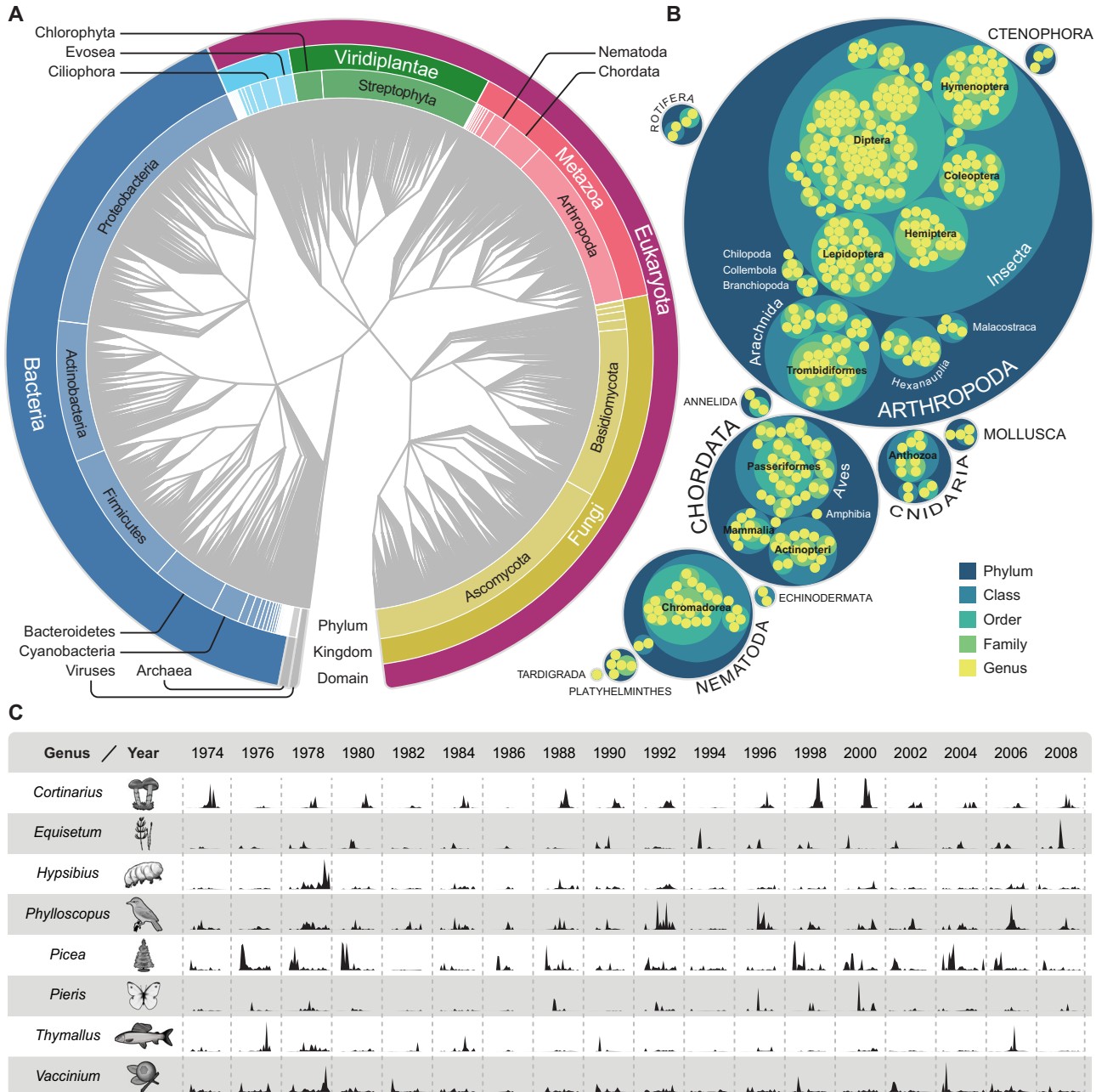

**Fig. 1 | Airborne eDNA provides reliable records of organisms across all domains of life. A** Taxonomic assignments of the 2739 genera detected in the air filters, according to the NCBI taxonomy. **B** A circular treemap showing all metazoan genera detected in the air filters ($n = 402$). Their taxonomic organization is visualized by nested circles, where the outermost circles represent phyla ($n = 11$, labels capitalized), and the innermost represent the individual genera. **C** Normalized read proportions (for each week, number of reads assigned to the genus divided by the total number of reads) from eight genera. Organism illustrations by Thomas Ågren.

lowland boreal forests and summer alpine pastures (Fig. 2D and E; Supplementary Methods). While the contribution from local (*ca.* < 50 km) animals are likely underestimated[42,48], these results show that long-distance transport contributes to the eDNA collected by the monitoring station. More precise estimates of geographic source regions require further research, such as size-stratified sequencing[24], but we show that airborne eDNA can reflect landscape-scale biodiversity.

### eDNA abundance indices correlate with traditional surveys

Field experiments in aquatic ecosystems support a complex but quantitative relationship between eDNA concentrations and animal abundance[30,31]. Direct measurements of eDNA concentrations are now

used in management and conservation, but inferring abundances from read counts remains controversial[49]. While some concerns are specific to metabarcoding[49] and not the shotgun sequencing used here, both methods produce catch-per-unit-effort (CPUE) data that are always affected by saturation[50]. As with traditional CPUE surveys, read counts can vary proportionally with abundance, but how often this holds true for empirical datasets is uncertain[51].

We compared abundance trends for eight bird genera from log-ratio transformed eDNA counts and traditional point-transect surveys (Fig. 3). Relative abundances from the traditional surveys explained 60% of the variation in eDNA relative abundances (adjusted $R^2 = 0.60$, $F(1, 38) = 8.6$, $p = 3.3 \times 10^{-9}$), suggesting that airborne eDNA can serve as a reliable proxy for conventional survey methods. Because airborne

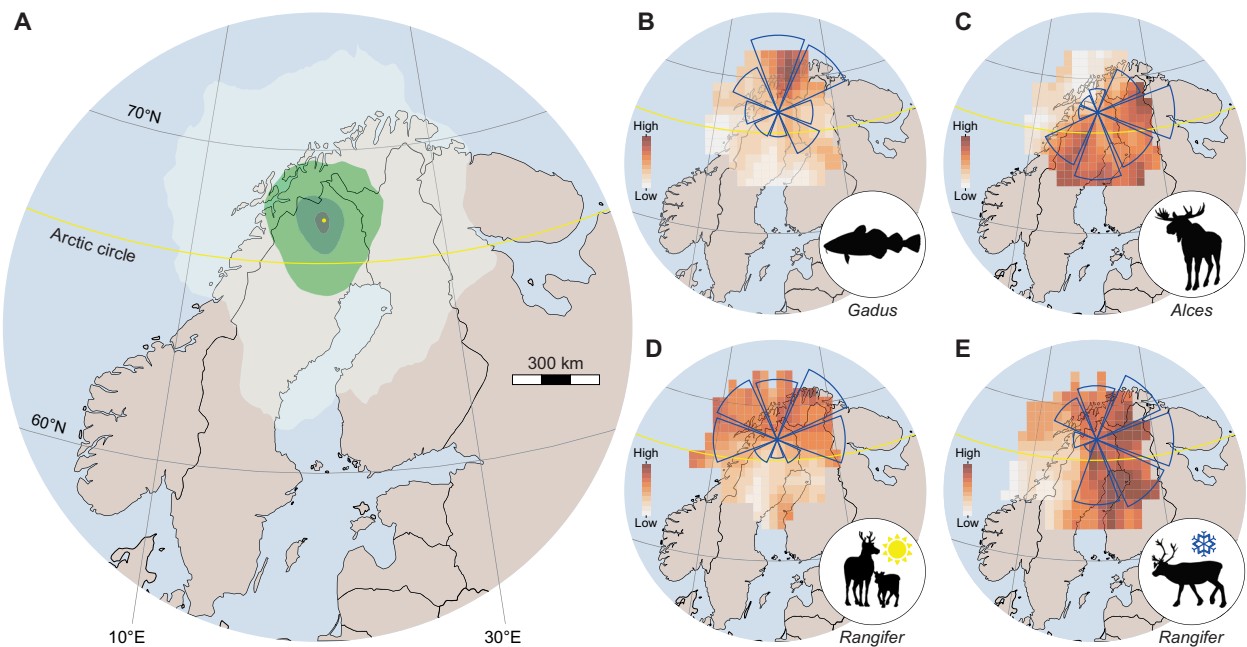

**Fig. 2 | Atmospheric modeling reveals potential geographic origins of eDNA.**
**A** Mean modeled origin densities of 22 μm particles in the catchment area during the study period. The differences in the intensity of green shading indicate a 10-fold difference in the number of particles captured from that area, assuming all areas have the same emission rate. The location of the monitoring station is indicated by a yellow dot. **B**–**E** Back-trajectory ensemble receptor models show the potential origins of eDNA from (**B**) cod (*Gadus*), (**C**) moose (*Alces*), and reindeer (*Rangifer*) during (**D**) summer weeks and (**E**) non-summer weeks. Color indicates signal strength, with darker shades showing stronger potential sources and lighter shades showing weaker ones. Blue wind roses display the relative strength of sources from each cardinal direction. Organism illustrations by Thomas Ågren and maps from Natural Earth.

eDNA can persist for decades under archival conditions, it offers a unique opportunity to extend and complement existing datasets to track long-term biodiversity trends. Longer time series across a broader range of taxa will enhance our understanding of how directional trends, environmental variation, and stochastic events shape populations over time.

## Airborne eDNA records seasonal and long-term changes in ecosystem composition

We identified seventeen groups of taxa with similar temporal trends through hierarchical clustering of pairwise log-ratio variances (Fig. 4A, Supplementary Data 6 and 8)[52]. Seasonal differences divided most organisms along higher taxonomic ranks: eDNA from eukaryotes generally peaked during a single season, whereas 88% of prokaryotic genera had their highest relative abundances during spring and autumn (Fig. 4A). A peak consistent with autumn sporulation distinguished most fungi from plants[36,53], and the early spring flowering of trees and dicotyledons separated them from the summer peaks of grasses[36] and mosses (Fig. 4A, B). The bimodal seasonality in prokaryotes, however, differed from prior evidence[53,54] and likely results from sequencing effects; that is, organisms with small genomes are most readily sampled when there is little competition in the sequencing pool.

In addition to phenology, coherent shifts in abundance can result from trophic interactions[6]. For example, the well-documented endosymbiosis between flies and *Rickettsiales* bacteria (Fig. 4B) and lichenized fungi and algae (Fig. 4A) can be detected from their strong temporal covariation (cluster C8 and C12, respectively). This suggests other clusters may reflect undiscovered interactions, such as between putatively endophytic *Venturiales* fungi[53] and pine (C6) or the rust fungi and grasses in cluster 4 (Fig. 4B)[55]. Shared temporal shifts may also indicate a shared response to environmental change[6] or aerosolization from a common substrate. A combination may explain the separation between groups of predominantly soil-dwelling (C1) *vs.*

endophytic fungi (C2)[53,56] and among bacteria associated with above-ground plant surfaces (C17)[53,57], animal hosts (C14)[58], and soils (C16)[53,56]. Abiotic conditions, direct trophic interactions, or aerosol emission fluxes that are in turn influenced by the environment[59] are all plausible hypotheses for the temporal variation found among micro-crustaceans, planktonic bacteria, and other aquatic microbes (C13).

We partitioned changes in the seventeen clusters into components explained by seasonality, longer-term trends, and environmental parameters using state space models. Ecosystems respond to shifting means, but changes in climatic variability and extremes are expected to be more mechanistically relevant to biota[60]. To capture some of this complexity, we compared the predictive skill of models using different combinations of latent trend structures and regression matrices, including 75 climatic covariates ($|\hat{\rho}| = 0.15$, $\sigma = 0.13$) and six comprising a null model of seasonal variation (Supplementary Methods). The best-performing models predicted 12 – 76% ($\hat{x} = 33\%$) of the variation in cluster relative abundance.

Climatic covariates improved forecasts for eight of the clusters, including all four dominated by plants and three of the four fungal clusters (Supplementary Data 11). Consistent with the timing of pollen and spore release in the boreal region, we found variables related to seasonal transitions to be reliable predictors of fungal and plant eDNA abundance (Supplementary Data 12 and 13). Fungi-dominated clusters generally increased with rain and snow, although eDNA from fungal endophytes (C2) was predictably lower up to 78 weeks after extreme rainfall events (Supplementary Data 12 and 13). Variables related to evapotranspiration were also selected by the models of some plant and fungal clusters, along with the bacterial genera in cluster 11 (Supplementary Data 12 and 13). In general, climatic covariables predicted weekly, seasonal, and cyclic variation but not multiannual or directional trends in relative abundances (Supplementary Data 12).

After removing the variation predicted by climatic covariates, we found robust evidence of long-term relative abundance trends in thirteen clusters (Fig. 5A, B and Supplementary Data 12). Most

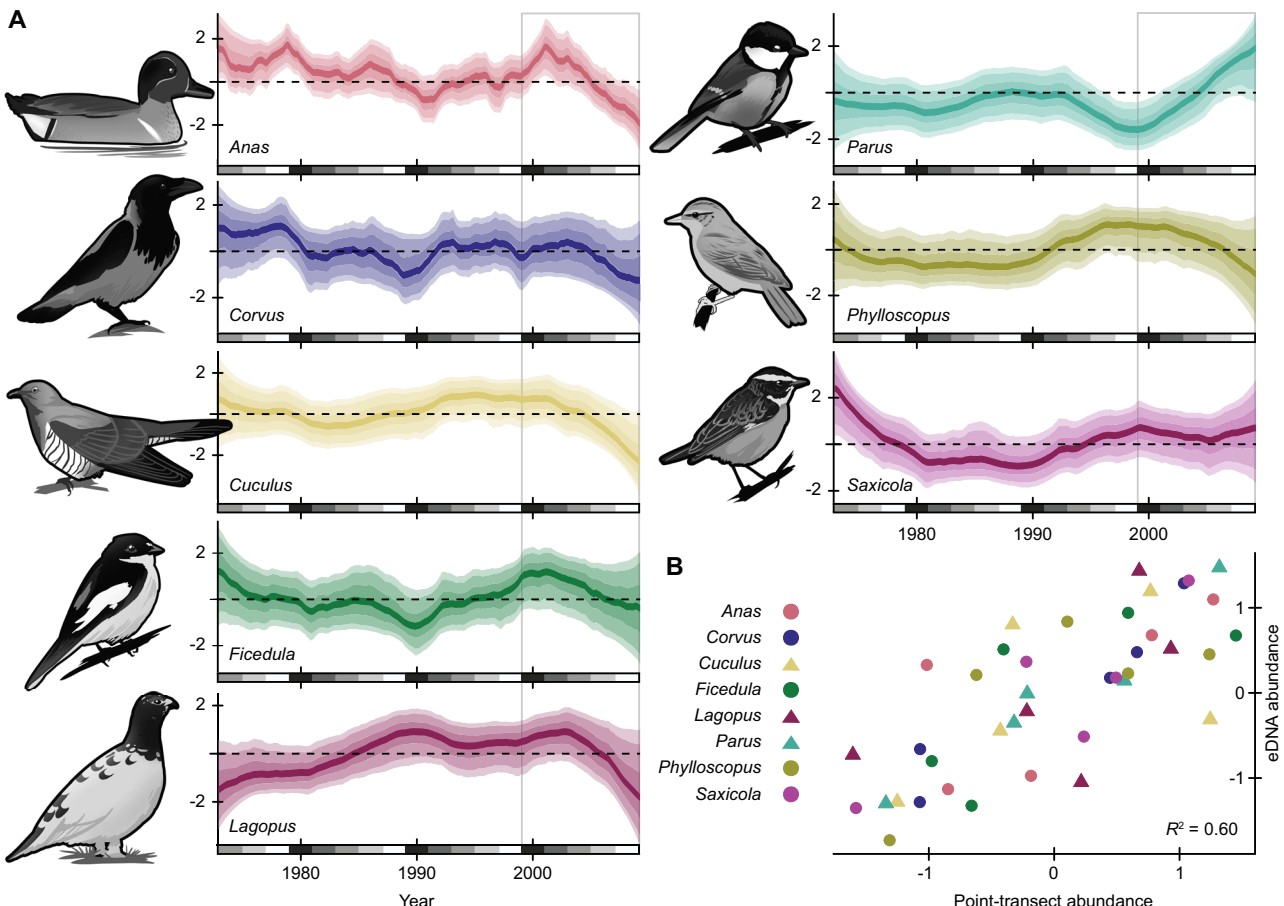

**Fig. 3 | Concordance of bird eDNA trends and monitoring data. A** Abundance trends reconstructed from eDNA using Bayesian state space models. Posterior median trends are indicated in bold, and shaded regions show the 95% credible interval. Gray boxes indicate the years overlapping with traditional point-transect survey data. **B** Relationship between modeled estimates of mean annual eDNA abundance and two-year moving average point-transect counts (adjusted $R^2 = 0.60$, $F(1, 38) = 8.6$, $p = 3.3 \times 10^{-9}$). Organism illustrations by Thomas Ågren.

conspicuously, the pine-dominated cluster (C6) increased from 40% of the entire community in the early years of the time series to 80% around 1994 followed by a gradual decline to 60% by 2008 (Fig. 5A). As these are relative abundance trends, a dramatic increase in one component forces declines among the others. However, the trends following this peak indicate a shift in community composition, rather than a saturation artefact driven by a transient spike in pine-associated eDNA. Nine clusters continued to decline even after 1994 and increases in relative abundance were unequally distributed among the other clusters (Fig. 5A, B). We also detected large relative abundance changes in some clusters in 2008, the end of our time series, which could indicate nascent trend reversals (Fig. 5B).

**Biodiversity loss from declines in forest taxa**

We used transformations of the Rényi entropies[61,62] to partition changes in biodiversity into evenness and distinctiveness components (Supplementary Methods). This framework extends the logic of Hill numbers[63] to relative entropy (β) and cross-entropy (γ) to obtain unified families of diversity indices. Higher α diversity indicates a more even relative abundance distribution in a week, whereas β increases as taxa are temporally structured. Changes in γ diversity occur through either, or both, of these components and indicate that biodiversity in a broad sense is unevenly distributed across time. Here, γ-diversity measures the weekly contribution to the total biodiversity over time.

Mean γ diversity declined between 1990 and 1994 (Fig. 5C), concurrently with the rapid increase of the pine cluster. Despite an increase from the mid-1990s, γ diversity averaged 35% lower (95% CI:

31–40%) between 2002–2008 than 1974–1988, a loss equivalent to *ca.* 31 effective taxa. Evenness decreased modestly but consistently over the same period, from 22 to 20 effective taxa (95% CI:17–30 to 15–27), although a steeper decline may have begun in 2008. This means the decline in γ diversity mostly resulted from a change in distinctiveness, with taxa more disproportionately abundant in 1974–1988 than in 2002–2008. Reducing the influence of rarer taxa ($q = 2, 3$) or restricting the analysis to different taxonomic subsets did not change this pattern of biodiversity loss (Supplementary Data 12).

Diversity metrics are not necessarily positively correlated with ecosystem health. Generalist and invasive taxa can increase diversity[64], even though their success often increases with environmental degradation[65]. We identified the taxonomic drivers of the diversity decline by comparing per-taxon γ contributions from 1978–1988 *vs.* 1994–2008. Consistent with the cluster trends, we found a large increase in the γ contribution of pine (two-sided Wilcoxon signed-rank test, Benjamini-Hochberg adjusted $p = 8.2 \times 10^{-14}$) and numerous declines in core taxa like birch (*Betula*; $p = 0.043$), spruce (*Picea*; $p = 0.66$), feathermoss (*Pleurozium*; $p = 2.5 \times 10^{-5}$), tree and ground-dwelling lichens, and wood-dwelling fungi (all $p < 0.001$), among other taxa with uncertain ecologies (Fig. 5D). These genera, and the species within them, occur in different habitats but are all directly affected by forest management[66–68].

Commercial forest management is extensive at the landscape scale (> 50 km): 1.5% of forests within *ca.* 300 km of the monitoring station were thinned or felled annually between 1986, the earliest year with reported data, and 2008. Forests in Fennoscandia are most

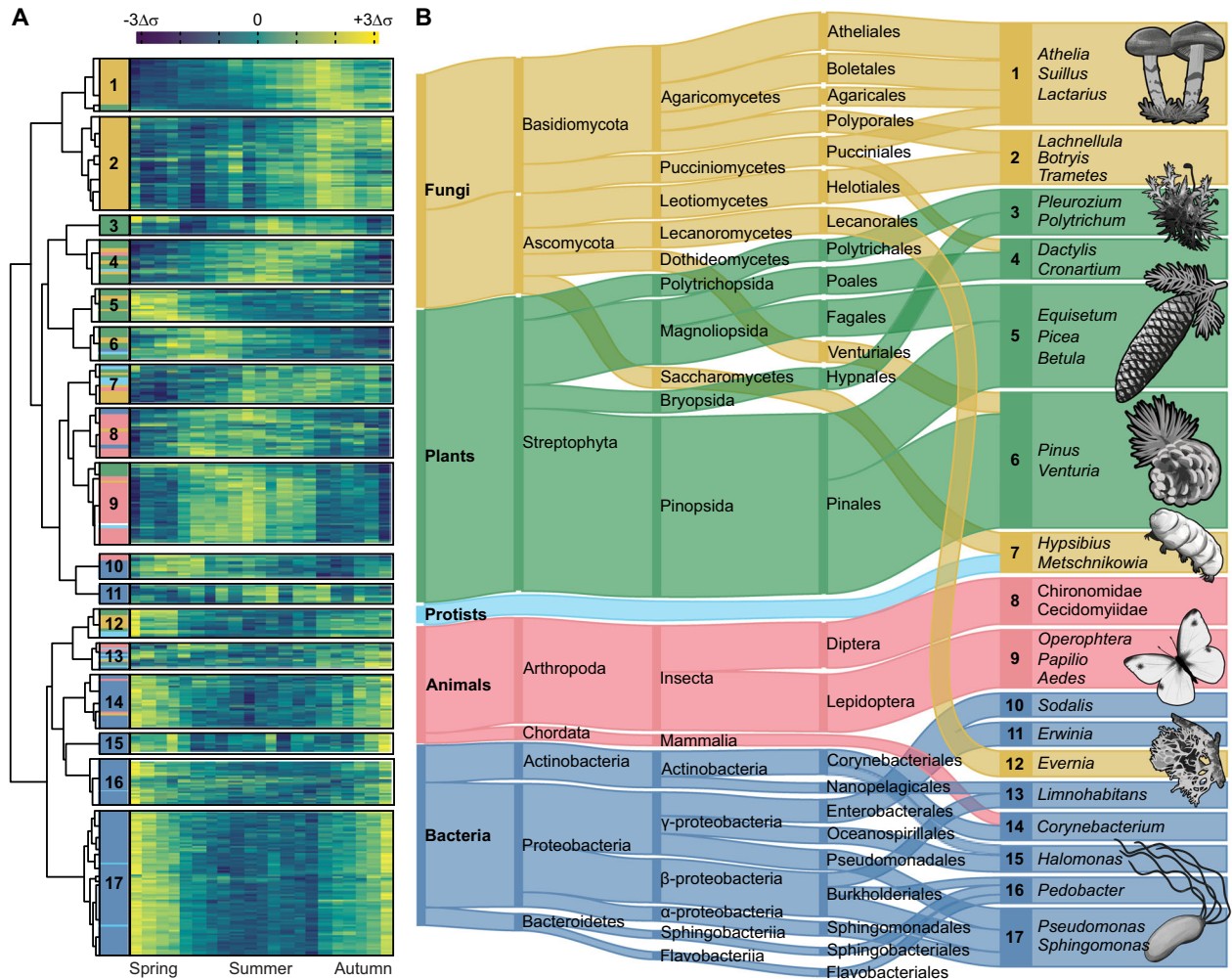

**Fig. 4 | Correlated shifts in airborne eDNA composition reveal temporal assemblages. A** Hierarchical clustering of the 2739 genera into 17 temporal clusters by their pairwise log-ratio covariances; stacked bars indicate kingdom membership and the heatmap shows median log-ratio transformed eDNA abundances for calendar weeks 21–41 (increasing from dark blue to bright yellow). Cluster sizes are approximately proportional to their taxon richness, but note that the largest clusters were reduced in size for display. **B** Taxonomic composition of the clusters from kingdom to order. Protists contain eukaryotes lacking a kingdom classification. Numbered boxes show representative genera for each cluster. Taxonomic groups comprising ≥ 5% of the dataset or a cluster are shown; ribbon and box heights are roughly proportional to rank relative abundances, but the lowest ranks are shown as ties for display. Organism illustrations by Thomas Ågren.

frequently clearcut, replanted with seedlings, and thinned multiple times before they are felled again. While effective for timber production, this silvicultural system has converted a structurally diverse landscape to a mosaic of monocultures. Between 1974 and 2008, primary forests in the region declined by >50% and more clearcuts occurred within 100 km of the monitoring station in the 1980s than any earlier period in the 20th century (Supplementary Methods). These forests were disproportionately replaced by pine, consistent with the long-term increase of pine-associated eDNA. On-the-ground management can create bioaerosol pulses that influence shorter-term eDNA trends[18] and the 1990–2000 pine maxima coincides with a period of extensive harvests and reforestation in the region (Supplementary Methods).

Population declines in taxa dependent on old forests, such as wood-dwelling fungi like *Porodaedalea* (Fig. 5D), are widely documented in Sweden[69,70]. Rare, specialist species like these are naturally vulnerable to environmental changes, but we also detected large γ declines in *Pleurozium* and *Trametes*, genera common in young, natural forests (Fig. 5D). Field-based studies have more recently emphasized the threats to these and other core genera posed by soil scarification[71], insufficient dead wood quantity or quality[38], habitat

fragmentation[72,73], or the altered light and moisture regimes from high planting densities and fire suppression[66,68]. Together, this suggests the largest change in airborne eDNA diversity resulted from commercial forest management across the landscape.

## Deep sequencing air can improve ecosystem surveillance capabilities

We demonstrate the ability of airborne eDNA to detect the contemporary presence of organisms across the tree of life, track shifts in ecosystem composition, and provide quantitative abundance indices. While this marks a notable improvement in the resolution and scope of eDNA biodiversity monitoring, amenability to reanalysis is a key benefit of our dataset. Most (76%) of our reads are unclassified, an unsurprising result given that only a tiny sliver of species have reference sequences[74]. With more extensive reference databases, future reanalysis of this dataset will continue to provide insights into biodiversity at multiple levels of organization.

Our study underscores the value of aerosol stations as serendipitous collectors of biodiversity data[10]. Our results suggest the high flow rates (500–1500 m³ h⁻¹) used in radionuclide detection also enable detection of even organisms that do not readily emit bioaerosols.

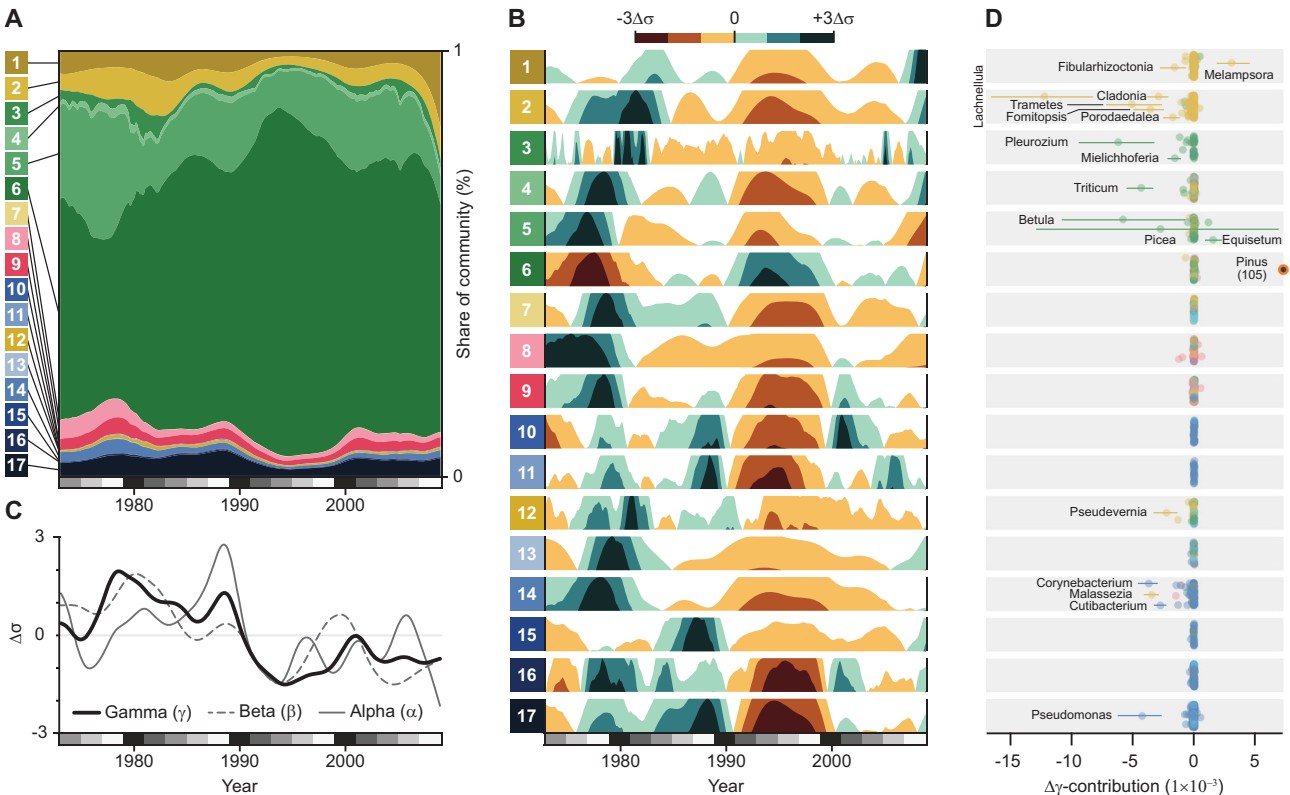

**Fig. 5 | Airborne eDNA records seasonal and long-term changes in ecosystem composition. A** Stacked area plot of posterior median relative abundances for the 17 clusters through time; total area represents the entire eDNA composition. **B** Horizon plots showing log-ratio transformed posterior median abundances, standardized within each cluster. Values show deviations from each cluster's long-term mean; color intensity indicates the magnitude of change. **C** Posterior median α, β, and γ diversities of the total composition. **D** Median difference in the γ- diversity contribution of each genus between 1974–1988 and 1994–2008, grouped by cluster. Error bars show two-sided 95 % confidence intervals from Wilcoxon signed-rank tests. The γ-diversity difference for *Pinus* (orange circle in cluster 6; Δγ = 0.105, 95% CI = 0.077, 0.132) falls outside the x-axis range. In (**A**, **B**), clusters dominated by fungi are shown in yellow hues, plants in green, metazoans in red, and prokaryotes in blue. In (**D**), individual genera follow the same color scheme, with protists shown in light blue.

Similarly to air quality networks[10], radionuclide stations operate worldwide under standardized protocols. Europe alone hosts more than 400 stations[75] and those surveilling for the Comprehensive Nuclear-Test-Ban Treaty Organization (CTBTO) are strategically positioned to maximize global coverage[76]. Airborne eDNA from these and other already operational networks may provide an unprecedented opportunity to reconstruct ecological history and detect ongoing changes almost in real-time.

## Methods
For additional details, see Supplementary Methods.

### Aerosol sampling
Air filters were collected once a week between 1974 and 2008 by the Swedish Defense Research Agency (FOI) to monitor radioisotopes in surface-level aerosols[77]. Samples were collected on glass fiber filters with a 0.2 μm pore size with a ~ 1000 m³ hr⁻¹ flow rate (mean sampling time 168 h). Filters prior to 1996 were Camfil type CS 5.0 (Camfil Svenska AB) and from then on HB5773 (Hollingsworth & Vose Company Ltd.). Filters are individually stored in airtight plastic containers in an archive.

**Laboratory methods.** Before DNA extraction, all samples were assigned a random number. DNA extraction, sequencing library construction and multiplexing in the sequencing was performed in batches according to the random sample IDs. We isolated DNA from filters installed during weeks with a mean temperature > 0 °C from even-numbered years between 1974 to 2008 using a protocol modified from

refs. 15,78,79. Previously, we found that filters from colder weeks did not reliably yield enough DNA for shotgun sequencing[15]. For each air filter, inside a sealed plastic bag, three Ø8 mm punches were collected in separate 2.0 mL screw cap tubes containing 1.0 g of 0.1 mm zirconia/silica beads and 0.5 g of 1.0 mm zirconia/silica beads (BioSpec). 1.0 mL lysis buffer (0.5 M EDTA, 0.5% Tween-20, 20 mg/mL Proteinase K) was added to each tube, briefly agitated for 10 s at 4.0 m/s and incubated at 37 °C overnight. Samples were agitated for the same duration and speed the next morning and then centrifuged 15 min at 16,000 × g. Supernatants (3 × 0.5 mL) were pooled in a 50 mL screw cap tube. This procedure was repeated twice, first with a 30 s, 5 m/s agitation followed by 15 min centrifugation and then by a 30 s, 6 m/s agitation followed by 5 min centrifugation. Prior to every agitation step, an additional 0.5 mL buffer (0.5 M EDTA, 0.5% Tween-20) was added to each filter punch. Supernatants were collected and added to their corresponding 50 mL tube to a total of approximately 4.5 mL. To each 50 mL tube, 8.8 volumes of binding buffer were added (5 M GuHCl, 40% Isopropanol, 90 mM NaAc, 0.05% Tween-20) followed by 10 s vortexing. Under vacuum, the solution was then passed through a Zymo-Spin IIICG column (Zymo Research). The column was washed once with 0.75 mL binding buffer and twice with 0.75 mL 80% Ethanol, before the DNA was eluted in 60 μL EB buffer. The eluted DNA was further cleaned using DNeasy PowerClean pro (Qiagen) and repaired using NEBNext FFPE DNA Repair Mix (New England Biolabs) according to the manufacturers' protocol. The final DNA concentrations were measured using Qubit Fluorometric Quantification and the Qubit 1X dsDNA HS Assay Kit (Thermo Fisher Scientific) (Supplementary Data 2). Libraries were prepared from isolates with a minimum of

~10 ng DNA at the Swedish National Genomics Infrastructure (SciLifeLab, SNP&SEQ, Uppsala) using the Thruplex DNA Seq kit (Takara) with 8 PCR cycles according to the manufacturer's protocol. Libraries were sequenced on Illumina NovaSeq 6000 S4 flow cells using 2 × 150 bp output (Illumina) (Supplementary Fig. 12). A blank control sample was included in each DNA extraction batch and in each library preparation batch.

**Bioinformatics.** We removed sequencing adapters using Cutadapt v. 2.0[80] and reads mapping to the human reference genome (hg19) with BBMap v. 38.69[81]. Taxonomic read classifications were made using a fork of Kraken 2 v. 2.0.8-beta[82] modified to report the number of distinct minimizers per taxon and a 4.2 terabyte custom reference database (Supplementary Data 3). We initially retained all classifications (i.e., --confidence 0 and --minimum-hit-groups 1) and used StringMeUp, a custom-built reclassification tool, and optimized the relationship between the fraction of reads classified to genera reported in the region (Supplementary Data 4) and the total fraction of classified reads at different levels of classification stringency. Based on these results (Supplementary Fig. 14), we enforced a minimum confidence score of 0.1 and a minimum of 10 hit groups for all classifications.

**Log-ratio transformations.** We calculated centered (CLR; Supplementary Methods Equation 1), isometric (ILR; Supplementary Methods Equations 2 and 3), and pivot log-ratios (PLR, a special case of ILR), as appropriate for a given analysis (Supplementary Methods 4.1). Prior to these transformations, we removed taxa with zero read counts for ≥ 2/3 of the weeks and imputed the remaining zeros using geometric Bayesian multiplicative replacement[83] as implemented by the cmultRepl function in the R package 'zCompositions' v. 1.4.0 176[84].

**Detrending.** We corrected for potential read length bias by using generalized linear models (GLMs) to model weekly ILR-transformed abundances as a function of their corresponding weekly mean read lengths with log, identity, and inverse link functions in the Python module 'statsmodels' v. 0.11.180[85]. We took the residuals from the best-fitting model for each genus, re-added the ILR-transformed means, and back-transformed them to relative abundances using the R package 'compositions' v. 2.0 679[86]. This detrended relative abundance matrix was then conditioned on air filter manufacturer, which changed in 1996, and human read count proportion in a redundancy analysis (RDA) with the R package 'vegan' v. 2.6 482[87]. The residuals from the RDA were then used as the basis for all subsequent analyses. Comparisons of relative abundances before and after detrending are shown in Supplementary Fig. 16.

**Post-classification taxon filtering with gradient boosting.** We calculated 31 statistics on PLR-transformed abundances based on the expectation that false positive genera should tend to have lower read counts, be detected rarely or unusually frequently, have distinct per read Kraken 2 classification quality metric profiles, and occur more frequently in lineages with more sequence data and/or larger genomes (Supplementary Methods 4.3.1). We also included one-hot encoded taxonomic kingdoms and the PLR-transformed weekly abundance of each genus as features.

Occurrence records from the Global Biodiversity Information Facility (GBIF) were used to curate a labeled training dataset. Genera with ≥ 4 observations within 40 km of the monitoring station between 1974–2008[29], along with humans, *Canis, Aedes*, and 33 bacterial genera from a comparable biome (NCBI Bioproject accession number PRJNA767205) were used as positive training data (n = 317). Genera with no reported occurrences within 5000 km of the monitoring station[88] and that did not share a family or lower taxonomic rank with any European taxa lacking a reference genome (Supplementary Methods 4.3.2) were used as negative training data (n = 379). Labeled

genera and feature data are provided in Supplementary Data 5, and their taxonomic composition summarized in Supplementary Table 5. A random subset (n = 91; 13%) were reserved as holdout test data.

We trained the GBM using the xgboost API for R[28] and performed a grid search with 5-fold cross-validation over 6561 hyperparameter combinations to identify the set achieving the smallest binary classification error rate. For the final model fit, we used the following hyperparameters: eta = 0.3, max_depth = 5, min_child_weight = 2, subsample = 0.7, colsample_bytree = 0.4, reg_alpha = 1e-05, gamma = 0.3, and reg_lambda = 1.5. Classification error rates were estimated from holdout test data (Supplementary Table 6). For the final classification, we retained genera with predicted probabilities ≥ 0.75, resulting in 2739 positive binary classifications.

**Classification validation with read alignments.** We mapped Kraken 2 classified reads to their respective sequences in the reference database using BBMap[81] v. 38.98 with the following parameters: pairedonly = t ambiguous = best killbadpairs = f minid = 0.97 and generated consensus sequences for the 100 regions with the highest read depth using the consensus utility in samtools[89] v. 1.20. Mapping results were assessed using the correlation coefficient between reference sequence length and the number of mapped reads. We used BLASTN as implemented in BLAST[90] v. 2.10.1 + to compare consensus sequences nt database and summarized results for best-scoring sequence pairs by taxonomic ranks. If BLASTN found BSPs with unrelated organisms, we generated consensus sequences for the next 100 regions by depth and repeated this process until no new potential contaminants were found.

**Aerosol dispersion models and catchment area estimation.** We used an adjoint version of PELLO[91], a random displacement Lagrangian particle model, to simulate particle transport backward in time from the monitoring station to their origin on the ground. We ran the model with numerical weather predictions from the ERA-5 dataset[92], covering 1980-2008 (excluding 1994, for which data was unavailable) using a 6 and 12 hour forecast step starting at 06:00 and 18:00, resulting in four forecast fields per day. The spatial domain of the weather data covered Europe, including the western part of Russia and Northern Africa. Aerosol dry and wet deposition were modeled, but no other aerosol dynamic processes were incorporated. As a source for the adjoint dispersion, we used particles with diameters of 5, 22, and 60 μm. The spatial domain of the release for the adjoint dispersion was defined with a horizontal domain of 30 × 30 m and a vertical domain stretching from 0–300 m, roughly corresponding to the planetary boundary layer in a neutral atmosphere.

We summarized the particle mass originating from eight cardinal directions and eleven distance classes (2, 5, 10, 20, 31, 50, 100, 180, 310, 520, and 860 km) for each sampled week from 1980 to 2008 (except 1994, Supplementary Data 1). We used these weekly sums for the 22 μm particle size as regression covariates in the time series analyses. Block bootstrapping with the R package "boot" v. 1.3-28[93,94] was employed to assess the range of particle dispersion and its associated uncertainty. Each bootstrap replicate consisted of 1000 resamples with a block size of four weeks, approximating a lag of one month. A 1000 draw Monte Carlo simulation was done to assess the 50% cumulative mass, using normalized bootstrapped weighted sums and their standard errors as input (Supplementary Fig. 4). Consistency of catchment area shape between particle sizes was tested with a two-way ANOVA with formula: dispersion ~ particle size × cardinal direction (Supplementary Table 1). Year-to-year variation in catchment area shape was evaluated with a linear mixed-effects model with scaled particle mass value as dependent variable, year and cardinal direction, including their interaction term as fixed effects, a random intercept for the year accounting for repeated measures and a first-order autoregressive term (Supplementary Table 2). Weekly variation was tested with a two-way ANOVA with formula: dispersion ~ week × particle size (Supplementary Table 3).

**Trajectory ensemble source receptor models of vertebrate eDNA**. Back-trajectories were calculated using HYSPLIT[95] using archived meteorology from the NCAR/NCEP reanalysis project[96]. Back-trajectories were started every 6 hours from the date the air filter was removed until the date it was installed with starting heights of 10, 100, 300, and 500 meters and followed for 24, 48, and 72 h. Simplified quantitative transport bias analysis[42,43] was used to compare modeled estimates of PLR-transformed eDNA abundances from *Alces*, *Gadus*, and *Rangifer* to the back-trajectories endpoints from HYSPLIT with the trajLevel function in the R package 'openAir' v. 2.18-2[97]. For comparison, we applied the same models to weekly measurements of cesium-137, a nuclear fission product, from air filters from the monitoring station between 1996 and 2006.

**Diversity metrics**. We partitioned the diversity observed in week *n* into alpha (α), beta (β), and gamma (γ) diversity components following the framework of refs. 61,62 (Supplementary Methods 6). We tested for significant differences in the weekly γ-diversity contributions from each genus in matched calendar weeks between the early and late years of the time series using the two-sided Wilcoxon rank sum test. We initially assessed the sensitivity of the results to the years used as the 'early' and 'late' periods using comparisons between '74-'80 *vs.* '02-'08, '74-'82 *vs.* '00-'08, '74-'84 *vs.* '98-'08, '74-'86 *vs.* '96-'08, and '74-'88 *vs.* '94-'08. We avoided comparisons including '90 and '92 because these years correspond to the temporary peak in *Pinus* abundance and the lowest γ-diversity. With the exception of *Picea*, we found no difference in the significance of Benjamini-Hochberg adjusted *p*-values (FDR = 0.05) or the direction of change for the genera with the largest differences in γ-diversity contributions (those in Fig. 5D). We therefore used '74-'88 *vs.* '94-'08 for the analysis. The median per-genus difference in γ-diversity contribution, 95% confidence intervals, and Benjamini-Hochberg adjusted *p*-values are given in Supplementary Data 9.

**Time series analyses**. We used Bayesian state-space models (SSMs) as implemented in the R package 'bsts' v. 0.9.9107[98,99] to fit structural time series models to PLR-transformed eDNA abundances (Supplementary Methods 4.1) and biodiversity metrics (Supplementary Methods 6). We compared time-dynamic specifications of the local linear trend and the integrated random walk models (Supplementary Methods 8) paired with regressor matrices comprising: (1) six trigonometric seasonality variables (Supplementary Methods Equation 11) ('base'), (2) these combined with 25 describing weekly catchment area variation ('particle'; Supplementary Methods 1.3), and (3) the seasonality variables combined 75 weather and climate-related variables (Supplementary Methods 7, Supplementary Data 10). Prior distribution selection is described in detail in Supplementary Methods 8.2.2 and summarized in Supplementary Table 8. We compared the predictive accuracy of models with different trend and regression specifications using the exact expected log pointwise predictive density (ELPD) estimated by leave future out cross validation[100,101]. If models could not be distinguished by ELPD differences, we considered the 'best' model to have fewer parameters and/or a smaller regressor matrix.

We used the F variance ratio between the first and last thirds of the time series, the magnitude and significance of autocorrelation in the first 42 lags, and Kolomogorov Smirnov's d to test for heteroscedasticity, serial dependence, and non-normality, respectively[102,103]. Convergence was evaluated by calculating effective sample sizes (ESS) for each parameter, Geweke's convergence diagnostic[104], Raftery and Lewis's diagnostic[105] with the R package 'coda' v. 0.19-4[106] and through visual inspection of parameter trace plots.

**Comparison with traditional monitoring data**. We compared counts from summer point-transect routes[107] with eDNA abundances for *Anas* (Anseriformes: Anatidae), *Corvus* (Passeriformes: Corvidae), *Cuculus* (Cuculiformes: Cuculidae), *Ficedula* (Passeriformes: Muscicapidae), *Lagopus* (Galiformes: Phasianidae), *Parus* (Passeriformes: Paridae), *Phylloscopus* (Passeriformes: Phylloscopidae), and *Saxicola* (Passeriformes: Muscicapidae). Four genera had multiple species included in the point-transect counts, which we summed and analyzed as a single genus.

We modeled summer point-transect routes using multivariate SSMs as implemented in the *R* package 'MARSS' *v*. 3.11.4[108]. For each genus, routes were treated as observers of the same latent population trend with different autoregressive errors (Supplementary Methods Equation 18)[109]. Models of PLR-transformed eDNA abundances were estimated as before (Supplementary Methods 8) and summarized as annual posterior means. We *z*-transformed the eDNA annual means and two-year moving averages of the point-transect trends and estimated their correlation with ordinary least squares regression.

## Reporting summary
Further information on research design is available in the Nature Portfolio Reporting Summary linked to this article.

## Data availability
The sequencing data generated in this study are deposited in the NCBI Sequence Read Archive (SRA) under accession code PRJNA808200. The processed relative abundance data are available in Supplementary Data 6. External datasets used are land cover data (Swedish National Land Cover Database, www.naturvardsverket.se/en/services-and-permits/maps-and-map-services/national-land-cover-database/), map vector data (Natural Earth, www.naturalearthdata.com/), weather data (Copernicus Climate Change Service, https://doi.org/10.24381/cds.e2161bac, National Centers for Environmental Prediction (NCEP) and National Center for Atmospheric Research (NCAR) Reanalysis project, psl.noaa.gov/data/gridded/reanalysis/, Swedish Meteorological and Hydrological Institute (SMHI), www.smhi.se/data/hitta-data-for-en-plats/ladda-ner-vaderobservationer, Climatology Lab, www.climatologylab.org/terraclimate.html, National Oceanic and Atmospheric Administration (NOAA) – Climate Prediction Center, www.cpc.ncep.noaa.gov, Expert Team on Climate Change Detection and Indices (ETCCDI), etccdi.pacificclimate.org/data.shtml), reference sequence data (National Center for Biotechnology Information (NCBI), www.ncbi.nlm.nih.gov/nucleotide/, accession numbers for all sequences used in the Kraken database are available at https://doi.org/10.5281/zenodo.17778887), species observational data (Swedish Species Observation System database, artportalen.se, Global Biodiversity Information Facility (GBIF), www.gbif.org, Swedish Bird Survey, www.fageltaxering.lu.se, Sámi Parliament of Sweden (Sámediggi), sametinget.se/renstatistik), and forestry data (The Swedish National Forest Inventory (NFI), www.slu.se/en/about-slu/organisation/departments/forest-resource-management/miljoanalys/nfi/, Swedish Forest Agency, www.skogsstyrelsen.se/laddanergeodata).

## Code availability
StringMeUp, a computer program developed in-house and used in the classification of the sequence data, and the Kraken 2 fork are both available under DOIs https://doi.org/10.5281/zenodo.17569636 and https://doi.org/10.5281/zenodo.17570001, respectively.

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

## Acknowledgements

We thank Catharina Söderström and Johan Kastlander (CBRN Defense and Security, Swedish Defense Research Agency) for providing access to the air filter archive, and Benedicte Albrectsen and Göran Englund for their feedback on previous versions of this manuscript. We also wish to thank five anonymous reviewers for constructive criticism. We acknowledge support from the Science for Life Laboratory and the National Genomics Infrastructure (NGI) for providing assistance in massive parallel sequencing. The computations were enabled by resources provided by the National Academic Infrastructure for Supercomputing in Sweden (NAISS) and the Swedish National Infrastructure for Computing (SNIC) at UPPMAX and HPC2N, partially funded by the Swedish Research Council through grant agreement nos. 2022-06725 and 2018-05973. Thomas Ågren provided the organism illustrations in Figs. 1–3. Modified Copernicus Climate Change Service information 2020 was used for the catchment area analysis. Neither the European Commission nor the European Center for Medium-Range Weather Forecasts (ECMWF) is responsible for any use that may be made of the Copernicus information or data it contains. This study was supported by Formas (grant agreement nos. 2016-01371: PS, MF; 2019-00579: P.S., T.B., and M.F.; 2021-02155: PS, MF; 2024-01990: P.S., T.B., M.F., and N.S.), together with grants from Vetenskapsrådet (2021-06283: P.S. and M.F.), SciLifeLab Biodiversity fund (NP00048: P.S., M.F., and T.B.), Kempe foundation (JCK-1919: P.S., M.F., and T.B.), Umeå University Industrial research school (P.S.) and Swedish Defense Research Agency (M.F.)

## Author contributions

P.S., M.F., T.B., and E.K. conceived and designed the study; E.K. and A.M.J. extracted DNA; DSv constructed the database and performed read classification; E.K., A.R.S., D.B., D.S.v. pre-processed the data; A.R.S. designed and implemented the machine learning approach; and H.G. constructed the particle models. A.R.S. and E.K. conducted most of the data analysis, with support from D.S.v., D.B., A.B.S., J.A.V., A.M., D.S.u., B.B., A.N., A.S., N.S., and P.A.E. E.K., A.R.S., D.S.v., B.B., P.S., and N.S. wrote the first draft of the manuscript. All authors contributed intellectual input and approved the final version

## Funding

## Competing interests

The Authors declare no competing interests.
