## [Transparent Peer Review file · Nature Communications]

Airborne eDNA captures three decades of ecosystem biodiversity

Corresponding Author: Dr Per Stenberg

Version 0:

Reviewer comments:

Reviewer #1

(Remarks to the Author)

I would like to acknowledge the extra hard work and objective reporting of the authors in robustly addressing my concerns relating to the previous lack of negative controls. Although the level of intensity of blanks now analysed associated with the *Drosophila* spike in method to enable sequencing is low, I am comfortable now that they have demonstrated that their controls were indeed populated with a very low number of background reads, that would be expected from a shotgun library preparation method. The insights derived from the qPCR work, in addition to the wealth of ecological signals that they report also add confidence to the data analysis and their findings. I also enjoyed reading about how the team addressed other reviewer's comments in relation to the additional ungulates in the datasets and note that these samples derive from ground level.

I would however encourage the team to be transparent in the main text about the blank samples and where to see the analyses, to avoid any unnecessary negative scrutiny from the broader community.

At present, they state that:

“While using laboratory methods robust to contamination (supplementary methods)”

I would at least suggest adding some words:

“While using laboratory methods robust to contamination and the analysis of lab and field controls (supplementary methods)”

I would be pleased to see the paper published and wish the authors well on future citations and synergies derived from their innovative work.

Reviewer #4

(Remarks to the Author)

I previously reviewed this manuscript and provided detailed comments regarding sampling site context, local positive controls, wildfire smoke as a tracer, cold-season filter usage, source-localization modeling, and interpretation of Pine Cluster 6. The authors have done a commendable job addressing these concerns and have significantly improved the manuscript.

Regarding sampling site transparency and airport influence, the revised manuscript now provides much more information about the sampling site, including Supp Fig 3 with site photographs (it's a small brown shack!). The authors have also explicitly acknowledged and discussed the proximity of the Kiruna airport, noting its low daily traffic (an average of seven arrivals per day) and arguing that any influence on airflows would have been modeled as sampling error in their state space time series models. While I am not fully convinced of the minimal influence assessment, I appreciate their transparency and willingness to clarify this point.

Regarding local positive controls such as reindeer and dogs, the authors provided thoughtful reasoning for why specific local examples (e.g., individual kennels) were not well suited to their analysis. Instead, they leveraged seasonal reindeer migration as a natural reference for source estimation and incorporated these data into their trajectory ensemble receptor models. This represents a creative and meaningful validation approach, and the manuscript has been updated with new results (highlighted in the right of Figure 2) and additional background on reindeer husbandry (interesting info!) (Supp Sect 9). With respect to dogs, the authors noted that individual kennels would be indistinguishable from the general dog population in the region and that dog walking commonly occurs along the access path to the monitoring station. While the authors chose not to pursue dog-specific analyses, they acknowledged the potential for future work (e.g., breed-level detection) and were transparent about this limitation.

Regarding wildfire smoke and chemical tracers, the authors examined relevant wildfire records noting that wildfires in northern Sweden were historically rare and generally small in spatial extent (with the exception of a 2006 outlier year). As such, they concluded that wildfires were not strong candidates for validating their atmospheric transport models. Importantly, this led the authors to leverage existing chemical data from the radionuclide monitoring station, particularly cesium-137 and beryllium-7 measurements from the same filters used for eDNA. The resulting source-region modeling of cesium-137 provides an innovative and rigorous approach to evaluate the accuracy and sensitivity of their atmospheric models. This addition strengthens the manuscript by integrating an independent tracer dataset.

Regarding cold-season filters, I had encouraged a closer look at filters from colder weeks as they could provide important context for long-range transport when local sources are minimized. The authors clarified that the archived filters represent a limited and valuable resource, with only ~5% of each filter available for eDNA analysis. They also noted that their previous metabarcoding pilot study (Karlsson et al. 2020) included winter weeks and revealed distinct seasonal signals (e.g., lichens and tree-associated fungi). In the current dataset, although winter weeks were not sequenced, the authors observed shifts in community composition (e.g., higher relative abundances of bacteria during early and late parts of the year). While this decision limits seasonal coverage, I appreciate the explanation and the consideration of trade-offs related to filter preservation and methodological differences between sequencing approaches.

Regarding source localization and particle size assumptions, the authors have expanded their discussion of limitations and clarified the objectives. It is my opinion that they now present a clearer distinction between their dispersion models (designed to illustrate how particle size influences transport potential) and their trajectory ensemble receptor models (TERMs) (agnostic to particle properties, instead using time series data to estimate likely source regions). This framing better contextualizes their approach as a qualitative tool for interpreting regional-to-landscape-scale trends rather than precise source attribution. In addition, the authors incorporated reindeer eDNA seasonal patterns into their trajectory models (very pleased the authors were excited about this suggestion!), generating results that align with known seasonal migration routes. They also integrated cesium-137 source modeling as an independent tracer validation. Overall, I find these revisions to be thoughtful and constructive, improving the robustness and interpretability of their source-localization framework.

Regarding Pine Cluster 6 and forestry practices, the authors expanded their discussion of Fennoscandian forestry, emphasizing the dominance of even-aged management and the widespread collection and chipping of residual biomass following clearcuts. This provides a plausible explanation for the observed “pulse” in airborne pine eDNA despite a concurrent decline in forest cover. However, I strongly encourage the authors to pursue direct microscopic observations of the archived filters, or at least formally request access for such analyses, as I had done previously. These observations are widely trusted in aerobiology and allergy communities and would add significant value and reach to the manuscript. If permission is denied, documenting that effort in the paper would still be worthwhile and transparent. Simply stating that access is “unlikely” without having requested it is not a convincing argument. I appreciate their acknowledgement of cellulose as an important bioaerosol with climate relevance and agree that future interdisciplinary efforts to integrate cellulose and other molecules into biodiversity monitoring would be valuable. Overall, I find their expanded silvicultural context helpful, but encourage additional effort to strengthen the link to established aerobiological methods.

In summary, the authors have addressed my previous concerns thoroughly and transparently, adding valuable context and improving the methodological clarity of the manuscript. Their integration of additional datasets (reindeer migration, cesium-137), expanded discussion of forestry practices, and clarification of modeling approaches make the work more robust and impactful. I still encourage the authors to aggressively pursue microscopic observations of pollen on the archived filters, as previously suggested, since this would strengthen the relevance of the study for aerobiology and allergy communities that rely on these trusted measurements. With that continued encouragement, I now find the manuscript suitable for publication, pending any minor editorial adjustments.

Reviewer #5

(Remarks to the Author)

Thank you very much for clarifying the utility of the log-transformation techniques adopted in the study. It would also be appreciated if the authors could amend the equations (1)—(3) and the notation throughout accordingly, by explicitly expressing the time variable t , for example. Readers will appreciate the authors' efforts and clarity.

The utility of PLR, in terms of applying it independently to each taxon (cluster), in the temporal context is still questionable, because each PLR transformation results in a different coordinate system, as the authors acknowledge. I feel that the parallel comparisons amongst those clusters' temporal patterns provide a little confusion, and I agree with the authors that

Fig. 5B could be misinterpreted.

Below are our point-by-point responses to the reviewer's comments in red text.

REVIEWERS' COMMENTS

Reviewer #1 (Remarks to the Author):

I would like to acknowledge the extra hard work and objective reporting of the authors in robustly addressing my concerns relating to the previous lack of negative controls. Although the level of intensity of blanks now analysed associated with the *Drosophila* spike in method to enable sequencing is low, I am comfortable now that they have demonstrated that their controls were indeed populated with a very low number of background reads, that would be expected from a shotgun library preparation method. The insights derived from the qPCR work, in addition to the wealth of ecological signals that they report also add confidence to the data analysis and their findings. I also enjoyed reading about how the team addressed other reviewer's comments in relation to the additional ungulates in the datasets and note that these samples derive from ground level.

I would however encourage the team to be transparent in the main text about the blank samples and where to see the analyses, to avoid any unnecessary negative scrutiny from the broader community.

At present, they state that:

“While using laboratory methods robust to contamination (supplementary methods)”

I would at least suggest adding some words:

“While using laboratory methods robust to contamination and the analysis of lab and field controls (supplementary methods)”

We agree with the reviewer and this sentence now reads:

“While using laboratory methods robust to contamination (see supplementary methods and analysis of lab and field controls in supplementary fig. 20), we also detected the platypus and other unlikely taxa after optimizing a lowest common ancestor read classifier²⁷ (supplementary methods).”

I would be pleased to see the paper published and wish the authors well on future citations and synergies derived from their innovative work.

We thank the reviewer for the constructive criticism of our manuscript that as a result is now significantly improved.

Reviewer #4 (Remarks to the Author):

I previously reviewed this manuscript and provided detailed comments regarding sampling site context, local positive controls, wildfire smoke as a tracer, cold-season filter usage, source-localization modeling, and interpretation of Pine Cluster 6. The authors have done a commendable job addressing these concerns and have significantly improved the manuscript.

Regarding sampling site transparency and airport influence, the revised manuscript now provides

much more information about the sampling site, including Supp Fig 3 with site photographs (it's a small brown shack!). The authors have also explicitly acknowledged and discussed the proximity of the Kiruna airport, noting its low daily traffic (an average of seven arrivals per day) and arguing that any influence on airflows would have been modeled as sampling error in their state space time series models. While I am not fully convinced of the minimal influence assessment, I appreciate their transparency and willingness to clarify this point.

Regarding local positive controls such as reindeer and dogs, the authors provided thoughtful reasoning for why specific local examples (e.g., individual kennels) were not well suited to their analysis. Instead, they leveraged seasonal reindeer migration as a natural reference for source estimation and incorporated these data into their trajectory ensemble receptor models. This represents a creative and meaningful validation approach, and the manuscript has been updated with new results (highlighted in the right of Figure 2) and additional background on reindeer husbandry (interesting info!) (Supp Sect 9). With respect to dogs, the authors noted that individual kennels would be indistinguishable from the general dog population in the region and that dog walking commonly occurs along the access path to the monitoring station. While the authors chose not to pursue dog-specific analyses, they acknowledged the potential for future work (e.g., breed-level detection) and were transparent about this limitation.

Regarding wildfire smoke and chemical tracers, the authors examined relevant wildfire records noting that wildfires in northern Sweden were historically rare and generally small in spatial extent (with the exception of a 2006 outlier year). As such, they concluded that wildfires were not strong candidates for validating their atmospheric transport models. Importantly, this led the authors to leverage existing chemical data from the radionuclide monitoring station, particularly cesium-137 and beryllium-7 measurements from the same filters used for eDNA. The resulting source-region modeling of cesium-137 provides an innovative and rigorous approach to evaluate the accuracy and sensitivity of their atmospheric models. This addition strengthens the manuscript by integrating an independent tracer dataset.

Regarding cold-season filters, I had encouraged a closer look at filters from colder weeks as they could provide important context for long-range transport when local sources are minimized. The authors clarified that the archived filters represent a limited and valuable resource, with only ~5% of each filter available for eDNA analysis. They also noted that their previous metabarcoding pilot study (Karlsson et al. 2020) included winter weeks and revealed distinct seasonal signals (e.g., lichens and tree-associated fungi). In the current dataset, although winter weeks were not sequenced, the authors observed shifts in community composition (e.g., higher relative abundances of bacteria during early and late parts of the year). While this decision limits seasonal coverage, I appreciate the explanation and the consideration of trade-offs related to filter preservation and methodological differences between sequencing approaches.

Regarding source localization and particle size assumptions, the authors have expanded their discussion of limitations and clarified the objectives. It is my opinion that they now present a clearer distinction between their dispersion models (designed to illustrate how particle size influences transport potential) and their trajectory ensemble receptor models (TERMs) (agnostic to particle properties, instead using time series data to estimate likely source regions). This framing better contextualizes their approach as a qualitative tool for interpreting regional-to-landscape-scale trends rather than precise source attribution. In addition, the authors incorporated reindeer eDNA seasonal patterns into their trajectory models (very pleased the authors were excited about this suggestion!),

generating results that align with known seasonal migration routes. They also integrated cesium-137 source modeling as an independent tracer validation. Overall, I find these revisions to be thoughtful and constructive, improving the robustness and interpretability of their source-localization framework.

Regarding Pine Cluster 6 and forestry practices, the authors expanded their discussion of Fennoscandian forestry, emphasizing the dominance of even-aged management and the widespread collection and chipping of residual biomass following clearcuts. This provides a plausible explanation for the observed “pulse” in airborne pine eDNA despite a concurrent decline in forest cover. However, I strongly encourage the authors to pursue direct microscopic observations of the archived filters, or at least formally request access for such analyses, as I had done previously. These observations are widely trusted in aerobiology and allergy communities and would add significant value and reach to the manuscript. If permission is denied, documenting that effort in the paper would still be worthwhile and transparent. Simply stating that access is “unlikely” without having requested it is not a convincing argument. I appreciate their acknowledgement of cellulose as an important bioaerosol with climate relevance and agree that future interdisciplinary efforts to integrate cellulose and other molecules into biodiversity monitoring would be valuable. Overall, I find their expanded silvicultural context helpful, but encourage additional effort to strengthen the link to established aerobiological methods.

Although we agree that microscopy analysis of the filters would be interesting and very informative, that analysis would require a significant effort and would be outside of the scope of the current manuscript. We thank the reviewer for this suggestion and will try to address this in future work.

In summary, the authors have addressed my previous concerns thoroughly and transparently, adding valuable context and improving the methodological clarity of the manuscript. Their integration of additional datasets (reindeer migration, cesium-137), expanded discussion of forestry practices, and clarification of modeling approaches make the work more robust and impactful. I still encourage the authors to aggressively pursue microscopic observations of pollen on the archived filters, as previously suggested, since this would strengthen the relevance of the study for aerobiology and allergy communities that rely on these trusted measurements. With that continued encouragement, I now find the manuscript suitable for publication, pending any minor editorial adjustments.

We thank the reviewer for the many suggestions for additional analyses and validation, and we agree that the added analyses have added great value to the manuscript.

Reviewer #5 (Remarks to the Author):

Thank you very much for clarifying the utility of the log-transformation techniques adopted in the study. It would also be appreciated if the authors could amend the equations (1)–(3) and the notation throughout accordingly, by explicitly expressing the time variable t , for example. Readers will appreciate the authors’ efforts and clarity.

We appreciate the insight and thorough revision of our methods. We have modified the equations (1)–(3) to indicate that the log-ratio transformations were performed on each weekly air filter sample independently. We believe these changes will improve the readability and clarity of the equations for the readers.

The utility of PLR, in terms of applying it independently to each taxon (cluster), in the temporal

context is still questionable, because each PLR transformation results in a different coordinate system, as the authors acknowledge. I feel that the parallel comparisons amongst those clusters' temporal patterns provide a little confusion, and I agree with the authors that Fig. 5B could be misinterpreted.

We feel the 5B panel is informative, but agree that with the given explanation in the figure legend it could be hard to interpret. We now explain both the A and B panel in more detail in the legend, so that it will be easier for the reader to interpret the results.

Figure 5 legend now reads:

“**A)** Stacked area plot of posterior median relative abundances for the 17 clusters through time; total area represents the entire eDNA composition. **B)** Horizon plots showing log-ratio transformed posterior median abundances, standardized within each cluster. Values show deviations from each cluster's long-term mean; colour intensity indicates the magnitude of change. [...]”

In addition, to make this figure easier to grasp, we changed:

The y-axis name of the A-panel from “Proportion” to “Share of community (%)”

The x-axis name of the D-panel from “Median diff. in γ contr.” To “ $\Delta\gamma$ -contribution”, as well as giving more information in the legend that now reads “**D)** Median difference in the γ -diversity contribution of each genus between 1974–1988 and 1994–2008, grouped by cluster. Error bars show two-sided 95 % confidence intervals from Wilcoxon signed-rank tests. The γ -diversity difference for Pinus (orange circle in cluster 6; $\Delta\gamma = 0.105$, 95% CI = 0.077, 0.132) falls outside the x-axis range.”

We also changed the colour shades of the clusters in the B-panel, so that they now match the A-panel.

We thank the reviewer for the valuable comments and believe that the above modifications now improve readability and interpretation of our results.